# Evaluating the Prognostic Value of the Triglyceride–Glucose Index in Different Populations: A Critical Analysis

**DOI:** 10.3390/nu17071124

**Published:** 2025-03-24

**Authors:** Antonio E. Pontiroli, Lucia La Sala, Elena Tagliabue, Graziella D’Arrigo, Stefano Ciardullo, Gianluca Perseghin, Giovanni Luigi Tripepi

**Affiliations:** 1Department of Health Sciences, Università Degli Studi di Milano, 20122 Milan, Italy; 2Department of Biomedical Sciences for Health, Università Degli Studi di Milano, 20133 Milan, Italy; 3IRCCS MultiMedica, 20138 Milan, Italy; elena.tagliabue@multimedica.it; 4Institute of Clinical Physiology of the National Research Council (IFC-CNR), 56124 Reggio Calabria, Italy; graziella.darrigo@cnr.it (G.D.); giovanni.tripepi@ifc.cnr.it (G.L.T.); 5Department of Medicine and Rehabilitation, Policlinico di Monza, 20900 Monza, Italy; stefano.ciardullo@unimib.it (S.C.); gianluca.perseghin@unimib.it (G.P.); 6Department of Medicine and Surgery, University of Milano Bicocca, 20126 Milan, Italy

**Keywords:** mortality, prognosis, obesity, age, sex, triglyceride, glucose, glucose tolerance, metabolic syndrome, Charlson comorbidity index, diabetes (T2DM), triglyceride–glucose-index

## Abstract

**Background/Objectives:** Recent studies have highlighted the Triglyceride–Glucose Index (TYG) as a significant risk factor for mortality and co-morbidities in various populations, including those with type 2 diabetes mellitus (T2DM) and cardiovascular diseases. However, its prognostic role in obese individuals remains less clear. **Methods**: Utilizing data from an obese cohort of 1359 subjects and from the 1999–2004 cycles of the National Health and Nutrition Examination Survey (NHANES) with 15,267 subjects, this study investigates the prognostic value of TYG and blood glucose in relation to age and sex and other factors such as metabolic syndrome, Charlson Comorbidity Index, T2DM and glucose tolerance, in predicting mortality among obese subjects. Over a median follow-up of about 13 years, 11.3% of the obese cohort and 20.6% of the NHANES cohort died. Our findings indicate that while TYG and blood glucose are significantly related to mortality, they offer only modest improvements over models incorporating age, sex, and other risk factors that showed a prognostic power of 76.1% and 86.0% in the respective cohorts. **Conclusions**: These results suggest that while TYG holds potential as a prognostic biomarker, its utility beyond established risk factors requires further validation in clinical settings.

## 1. Introduction

Several papers have appeared recently on the role of the Triglyceride–Glucose Index (TYG). Originally introduced as a low-cost surrogate index of insulin resistance [1,2], TYG is a risk factor for mortality and co-morbidities in the general population, in diabetes mellitus (T2DM), and in patients with cardiovascular diseases [2,3,4]; only in 2024, 630 papers have appeared in PubMed dealing with TYG [5]. 

Insulin resistance (IR) is a pathophysiological condition distinguished by the diminished effectiveness of insulin in promoting glucose uptake and utilization. The Triglyceride Glucose (TYG) index, derived from fasting triglyceride (TG) and fasting plasma glucose (FPG) levels, has become a surrogate marker of IR. The TYG is calculated using the formula [1,2]:ln [*fasting triglycerides* (mg/dL) × *fasting glucoses* (mg/dL)]/2

This allows simultaneous tracking of changes in triglyceride and glucose levels. Several studies have demonstrated that the TYG correlates with the advancement of metabolic disorders. For instance, in a large National Health and Nutrition Examination Survey (NHANES) 1999–2018 cohort of T2DM patients, TYG enhanced the risk of all-cause mortality in patients aged <65 years, but not in older patients [6]. However, a recent study performed in an Iranian cohort of the general population suggested that the impact of TYG on mortality is limited to T2DM subjects [7]. Older studies on the prognostic value of TYG usually did not compare the results obtained with other possible prognostic indexes (the majority) or showed a similar value [8,9,10].

The role of TYG in obese subjects is less defined; the only available data indicated that TYG was independently associated with harmful cardiovascular events in young and middle-aged obese US populations [11]. Quite recently, it was shown that TYG, just like blood glucose and metabolic syndrome, is associated with the deterioration of glucose tolerance and with all-cause mortality in obesity [12,13]. 

Especially with TYG, but also with other suggested prognostic factors (be it the metabolic syndrome [14], the Charlson Index [15], Glucose tolerance (GT), or blood glucose levels), one might wonder how these prognostic factors improve the value of classical and universal risk factors, such as age and sex. To be adopted in everyday clinical practice, a candidate prognostic biomarker should provide prognostic information that exceeds what is offered by well-validated and simple risk prediction rules. For instance, current guidelines recommend considering echocardiography to refine cardiovascular (CV) risk assessment in hypertensive patients, and the measurement of left ventricular mass index (LVMI) is considered an important independent risk factor for mortality and heart failure [16]. However, LVMI did not meaningfully improve risk prediction for these two conditions when added to clinical models [17,18]. 

The aim of our study was to analyze, in obese subjects [13], the additive role of these prognostic factors (TYG and blood glucose) over more classical and universal risk factors such as age and female sex, plus one of the additional risk factors [metabolic syndrome, T2DM, the Charlson Comorbidity Index, GT, arterial hypertension (AH), and coronary heart disease (CHD)]. To reduce the possible drawbacks of an analysis based on a single cohort, we chose to analyze some of the data in the general population (data obtained from the 1999–2004 cycles of the NHANES [19]) and in a subgroup of obese subjects from the same cohort.

## 2. Materials and Methods

### 2.1. Cohorts in the Study

The first cohort includes obese patients receiving routine medical treatment [13]; subjects undergoing bariatric surgery (BS) were excluded from the study since BS can reduce mortality [20,21]. Therefore, we analyzed a cohort of 1, 359 obese subjects (371 men and 988 women, aged 44.1 ± 12.6 years, with a body mass index (BMI) of 39.9 ± 5.2 kg/m^2^, followed by a median period of 13.9 years). The second study cohort includes subjects from the 1999–2004 cycles of the NHANES, [19]; it is made of 15,267 subjects (7389 men and 7878 women, aged 47.1 ± 19.7 years, with a BMI of 28.4 ± 6.5 kg/m^2^, followed for a median period of 13.3 years). A sub-cohort of the same cohort [19] with BMI > 30 kg/m^2^ was also evaluated. In both cohorts, blood glucose and TYG were expressed as quartiles of distribution (blood glucose quartiles (BGQ) and TYG quartiles (TYGQ), respectively). This study represents a second analysis of studies approved by Ethics Committees. The first study protocol [13] was approved by local Ethics Committees in 2015 (Coordinating Center: Ospedale San Paolo, Comitato Etico Interaziendale di Milano Area A, official approval SC: 2015 ST 125), and written informed consent was obtained from all adult participants. The second study protocol [19] was originally approved by the Centers for Disease Control and Prevention Research Ethics Review Board, and written informed consent was obtained from all adult participants. The present analysis was deemed exempt by the Institutional Review Board at our institutions, as the dataset used in the analysis was completely re-identified. 

### 2.2. Methods and Procedures

Clinical and laboratory characteristics of subjects of the two cohorts under study are shown in Table 1. 

### 2.3. Statistical Analysis

In obese patients of Cohort 1, the prognostic value for mortality BGQ and TYGQ was investigated in analyses of increasing complexity, starting with a basic model, including age and sex, and one of the classical risk factors at a time (T2DM, Charlson Comorbidity Index, Metabolic Syndrome, GT, AH, and CHD). In Cox analyses, data were expressed as the hazard ratio (HI), 95% confidence interval (CI), and *p*-value. The predictive accuracy of each prognostic model was assessed by calculating the Harrell C-index, ranging from 50% to 100%. A Harrell’s C index of 50% indicates no discrimination ability of the model being tested, whereas a Harrell’s C of 100% indicates perfect discrimination [22]. Some models were analyzed also in the general population. Data analysis was performed by STATA 17 (Stata Corporation, College Station, TX, USA) for MacIntosh. 

## 3. Results

The two cohorts were not intended for direct comparisons and differed in size, sex ratio, BMI, and frequency of cardiovascular diseases; other subtle differences were observed in the frequency of diabetes, metabolic syndrome, and overall mortality, despite similar durations of follow-up and similar frequencies of other medical and metabolic conditions.

Over a median follow-up of 13.9 years, 154 out of 1359 obese subjects of Cohort 1 died (11.3%); over a median follow-up of 13.3 years, 3136 out of 15,267 subjects of Cohort 2 died (20.6%). In cohort 1, a statistical model, including age, sex, and T2DM (Table 2—Basic model), provided a 76.1% prognostic power for predicting mortality. In this model, age, female sex, and T2DM were found to be significantly related to death (always *p* < 0.001). In cohort 2, a statistical model, including age, sex, and T2DM (basic model), provided an 86.0% prognostic power for predicting mortality. In this model, both age, female sex, and T2DM were found to be significantly related to death (always *p* < 0.001). In either cohort, the inclusion of BGQ (Model 1) or TYGQ (Model 2), or both (Model 3), into Model 1 did not materially increase the prognostic power of the same model (HC index ranging from 76.1% to 76.4%, and from 86.0% to 86.1%). Appendix A reports the prognostic value of risk factors at univariable Cox regression analyses. This analysis revealed that all risk factors resulted in being significantly related to mortality with a discriminator power (Harrell’C index) ranging from 56.2% to 73.7%. 

Appendix A reports the prognostic value of other risk factors for cohort 1 (A = GT-based model; B = metabolic syndrome-based model; C = AH-based model; D = CHD-based model; E = Charlson Comorbidity index- based model; F = BG based models) and for cohort 2 (G = diabetes analysis restricted to subjects with BMI > 30 kg/m^2^). In all models and in either cohort, the inclusion of BGQ (Model 1), TYGQ (Model 2), or both (Model 3) into the Basic Model did not meaningfully increase the prognostic power of the same model, the maximum increase being 1.32% (HC index ranging from 76.3% to 76.6%, and from 83.6% to 83.7%, respectively). When BG was entered instead of BGQ, and TYG instead of TYGQ, the significance of the models did never change. In synthesis, the predictive power of TYG did not significantly improve over traditional models.

## 4. Discussion

In this study, in a large cohort of obese patients (cohort 1) and in an even larger cohort of the general population (cohort 2), multivariable models of various complexities showed that BGQ and TYGQ did not meaningfully improve risk prediction of mortality when added to a prediction model that included only age, sex, and other classical risk factors. 

Two recent meta-analyses revealed significant associations between TYG and various health outcomes, suggesting that TYG is a promising biomarker for screening and predicting various medical conditions [23,24]. However, the heterogeneity and methodological quality of the included studies suggest the need for further high-quality research to confirm these findings and refine the clinical utility of the TYG [20,21]. Only a few studies explored the value of TYG in obesity [11,12,13,25,26], but no one has analyzed the additive role of TYG beyond more classical and universal risk factors such as age and female sex, metabolic syndrome, the Charlson Comorbidity Index, diabetes, glucose tolerance, and blood glucose levels.

In this study, initial univariable Cox regression analyses indicated that all risk factors were significantly related to mortality, with Harrell’s C-index values ranging from 53% to 74%, indicating varying levels of discriminatory power. We found that in obese patients and in the general population, age and sex, diabetes, glucose tolerance, metabolic syndrome, the Charlson Comorbidity Index, arterial hypertension, or CHD are significant prognostic factors for mortality. However, while age and sex are coherently significant risk factors, other risk factors show a significant prognostic value when introduced alone, and their effect reduces or vanishes (cohort 1), or is unaffected (cohort 2) when introduced in the model together with BGQ or TYGQ. For cohort 1, similar aspects were found for diabetes, glucose tolerance, metabolic syndrome, the Charlson Comorbidity Index, arterial hypertension, or CHD; for cohort 2, similar aspects were found when the sample of subjects was reduced to subjects with BMI > 30 kg/m^2^. Even though BGQ and TYGQ were significant risk factors when introduced alone, neither BGQ nor TYGQ were of significant value when introduced with other risk factors, either individually or together. Of note, even when statistically significant, the additive role of TYGQ or BGQ was of very limited value, with a maximum percentage increase in the Harrell index of around 1.3%, meaning that none of the factors investigated materially increase the prognostic value of basic models. Of note, a recently published paper [27] investigated the predictive value of adding TYG to Cox regression models, including traditional risk factors, and found that the increase in Harrell’s C index was 0.008, which is not different from our data of 0.012. Reasons why neither TYGQ nor BGQ were of significant value as predictors of mortality when added to other traditional risk factors are a matter of speculation. For instance, even though TYG is intuitively more complex than BGQ because of the contemporaneous evaluation of blood glucose and triglycerides, their contribution to other metabolic risk factors like diabetes, glucose tolerance, and metabolic syndrome, is likely to be an overvaluation of aspects already present in the same conditions. A similar possibility applies to the Charlson Comorbidity Index, which is built on the number of diseases of the individual patient. In addition, since TYG was originally introduced as a low-cost surrogate index of insulin resistance [1,2], similar considerations can apply to arterial hypertension, which is generally characterized by insulin resistance [28,29]. A better prognostic index might be represented by TYG-derived indexes, such as TYG-BMI, TYG-waist circumference, TYG-height, or TYG-waist-hip ratio, which were not analyzed in this study and appear to be more complete indexes [8,9,10,30,31,32,33,34,35,36,37]. 

This suggests that while various health and biochemical parameters are individually associated with mortality, age, and sex are the most robust predictors in this cohort. Furthermore, this study, for the first time, demonstrates that although TYG and blood glucose were significantly related to survival in obese patients (i.e., the corresponding hazard ratios are statistically significant), the same biomarkers were not useful for prognostic purposes. This is because the hazard ratio is an index well suited to etiological research but does not, per se, provide information on the prognostic accuracy of a given exposure [38]. Thus, while blood glucose and TYG remain fundamental treatment targets in obese patients, measuring these two factors solely for risk stratification is not justifiable in this condition. In conclusion, other studies are required to state if TYG, as well as blood glucose, can be used as prognostic indexes for mortality in obese subjects and in the general population. The present findings suggest that both TYG and blood glucose, if any, have a very limited role.

## Figures and Tables

**Table 1 nutrients-17-01124-t001:** Details of the subjects in the two cohorts. Mean ± SD.

Obese Cohort [13]		General Population Cohort [19]	
Number (sex M/F) (% men)	1359 (371/988) (27.3%)	Number (sex M/F) (% men)	15,267 (7389/7878) (48.4%)
Age (years)	44.1 ± 12.6	Age (years)	47.1 ± 19.7
Body mass index (BMI, kg/m^2^)	39.9 ± 5.2	Body mass index (BMI, kg/m^2^)	28.4 ± 6.5
Median duration of follow-up (years)	13.9	Median duration of follow-up (years)	13.3
BG (mg/dL)	118.1 ± 47.2	BG (mg/dL)	105.4 ± 35.5
TYG *	8.9 ± 0.7	TYG *	8.7 ± 0.7
Cholesterol (mg/dL)	212.8 ± 66.4	Cholesterol (mg/dL)	196.8 ± 43.2
HDL-cholesterol (mg/dL)	50.0 ± 13.6	HDL-cholesterol (mg/dL)	53.4 ± 15.9
LDL-cholesterol (mg/dL)	136.5 ± 64.3	LDL-cholesterol (mg/dL)	117.6 ± 36.2
Triglycerides (mg/dL)	159.5 ± 133.2	Triglycerides (mg/dL)	140.7 ± 121.2
AST (U/L)	25.9 ± 13.8	AST (U/L)	25.6 ± 0.2
ALT(U/L)	35.2 ± 24.0	ALT(U/L)	28.9 ± 0.4
Creatinine (mg/dL)	0.8 ± 0.2	Creatinine (mg/dL)	0.9 ± 0.42
Arterial hypertension (%)	425 (31.3%)	Arterial hypertension (%)	5487 (35.9%)
Type 2 diabetes (%)	131 (9.6%)	Type 2 diabetes (%)	1863 (12.2%)
Metabolic syndrome (%)	717 (52.8%)	Metabolic syndrome (%)	4990 (35.1%)
Cardiovascular disease (%)	51 (3.8%)	Cardiovascular disease (%)	1130 (9.3%)
All-cause mortality (sex M/F) (%)	154 (59/95) (11.3%)	All-cause mortality (sex M/F) (%)	3136 (1737/1399) (20.6%)

BG = blood glucose; TYG = triglyceride–glucose-index; ALT = alanine transaminase; AST = aspartate transaminase; * TYG = ln [triglycerides (mg/dL) × blood glucose (mg/dL)/2].

**Table 2 nutrients-17-01124-t002:** Cox regression analysis and Harrell’C index.

**A. Obese Cohort.**				
	**Basic Model**	**Model 1**	**Model 2**	**Model 3**
**Variables (Units of Increase)**	**HR (95% C.I.),** ***p* Value**	**HR (95% C.I.),** ***p* Value**	**HR (95% C.I.),** ***p* Value**	**HR (95% C.I.),** ***p* Value**
Age (years)	1.078 (1.060–1.096), *p* < 0.001	1.075 (1.057–1.094), *p* < 0.001	1.077 (1.059–1.095), *p* < 0.001	1.076 (1.058–1.094). *p* < 0.001
Sex (females versus males)	0.535 (0.386–0.744), *p* < 0.001	0.539 (0.388–0.749), *p* < 0.001	0.561 (0.403–0.781), *p* = 0.001	0.560 (0.402–0.779), *p* = 0.001
Diabetes	1.817 (1.314–2.512), *p* < 0.001	1.341 (0.802–2.244), *p* = 0.264	1.473 (1.009–2.150), *p* = 0.045	1.259 (0.747–2.124), *p* = 0.387
BGQ		1.213 (0.935–1.574), *p* = 0.146		1.126 (0.855–1.483), *p* = 0.398
TYGQ			1.210 (1.006–1.456), *p* = 0.043	1.176 (0.967–1.430), *p* = 0.103
Harrell’C index	76.1%	76.3%	76.3%	76.4%
**B. General Population Cohort.**				
	**Basic Model**	**Model 1**	**Model 2**	**Model 3**
**Variables (Units of Increase)**	**HR (95% C.I.),** ***p*-** **Value**	**HR (95% C.I.),** ***p*-** **Value**	**HR (95% C.I.),** ***p*-** **Value**	**HR (95% C.I.),** ***p*-** **Value**
Age (years)	1.094 (1.091–1.097), *p* < 0.001	1.094 (1.091–1.097), *p* < 0.001	1.094 (1.091–1.097), *p* < 0.001	1.094 (1.091–1.097), *p* < 0.001
Sex (females versus males)	0.662 (0.617–0.711), *p* < 0.001	0.661 (0.615–0.709), *p* < 0.001	0.663 (0.617–0.711), *p* < 0.001	0.661 (0.615–0.710), *p* < 0.001
Diabetes	1.487 (1.371–1.612), *p* < 0.001	1.505 (1.377–1.645), *p* < 0.001	1.501 (1.379–1.633), *p* < 0.001	1.510 (1.381–1.652), *p* < 0.001
BGQ		0.988 (0.952–1.025), *p* = 0.528		0.992 (0.954–1.032), *p* = 0.689
TYGQ			0.985 (0.947–1.024), *p* = 0.447	0.988 (0.948–1.029), *p* = 0.560
Harrell’C index	86.0%	86.0%	86.0%	86.1%

BGQ = blood glucose quartiles; TYGQ = triglyceride-glucose-index Quartiles.

## Data Availability

The research data will be available on request to the authors. The ZENODO platform is used for publicly archived datasets.

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
