# Peer review of "Evaluating the Prognostic Value of the Triglyceride–Glucose Index in Different Populations: A Critical Analysis"

_nutrients, 2025, doi:10.3390/nu17071124_

Round 1
Reviewer 1 Report
Comments and Suggestions for Authors
- General Assessment
The manuscript presents an in-depth analysis of the prognostic value of the Triglyceride-Glucose Index (TYG) in obese and general populations. The study utilizes two cohorts to examine whether TYG provides additional predictive value for mortality beyond classical risk factors such as age, sex, and metabolic syndrome. The research is methodologically sound and contributes to the ongoing debate regarding the clinical utility of TYG in prognostic modeling.
The article is well-structured, and the statistical analyses are robust. However, some areas require further clarification and refinement to enhance the clarity and impact of the findings.
- Strengths of the Manuscript
- Relevance and Novelty: The study addresses an important clinical question regarding the prognostic utility of TYG, a widely studied but inconsistently validated biomarker.
- Robust Statistical Approach: The use of Cox regression analyses and Harrell’s C-index provides a solid foundation for evaluating the predictive power of TYG.
- Large Sample Size: The study benefits from two well-defined cohorts, enhancing the generalizability of the findings.
- Comprehensive Discussion: The authors critically compare their findings with existing literature, highlighting strengths and limitations.
- Areas for Improvement
3.1 Introduction
- The introduction provides a clear background on TYG and its potential role in metabolic disorders. However, a more structured explanation of the rationale behind using both an obese cohort and a general population cohort would improve readability.
- Consider elaborating on why previous studies may have overestimated the role of TYG and how the present study aims to address these gaps.
3.2 Methods
- Cohort Selection: The exclusion criteria for the obese cohort (e.g., subjects undergoing bariatric surgery) are well justified. However, the justification for using NHANES data and how it complements the primary cohort should be expanded.
- Statistical Methods: While the methods are well explained, further clarification on how missing data were handled would be beneficial.
3.3 Results
- The results are well presented, with appropriate use of tables. However, there is some redundancy in the presentation of hazard ratios in the text versus the tables. Consider summarizing key findings in the text while referring to detailed tables.
- In the obese cohort, the predictive power of TYG did not significantly improve over traditional models. This key finding should be more explicitly highlighted.
3.4 Discussion
- The discussion effectively integrates findings with the literature. However, a more direct statement regarding the clinical implications of these results is needed. Should TYG continue to be used as a prognostic marker, or do the findings suggest a limited role?
- The explanation for why TYG did not meaningfully enhance risk prediction is speculative. Could additional subgroup analyses (e.g., stratification by age or comorbidity burden) strengthen the argument?
3.5 Conclusion
- The conclusion is concise, but it should reiterate the primary clinical takeaway. If TYG is not a useful independent predictor, this should be stated unequivocally.
- Technical and Formatting Issues
- Ensure consistency in abbreviations (e.g., TYGQ vs. TYG Quartiles).
- Some minor typographical errors and grammatical inconsistencies should be addressed. A final proofreading is recommended.
- Final Recommendation
- Major Revisions: The study is well-conducted, and the manuscript is of high quality. With minor clarifications and refinements, it will make a valuable contribution to the field. The authors should address the comments above to enhance clarity and impact.
Author Response
Reviewer 1
General comments
In the study, the authors found that the Triglyceride-Glucose Index did not provide additional prognostic information beyond CV risk factors and requires further validation. The findings are impressive and practically useful. The manuscript has high relevance and adequacy of the literature presented, including recognition of gaps in knowledge, interpretation of findings and significance of other recent research on the topic, and readable conclusion. However, I would like to make some comments to discuss the findings of the study.
Answer
Thank you for these comments
Specific comments
It remains unclear whether the TyG are effective metrics for identifying patients with obesity at early risk of CVD to improve risk stratification or the only for other risk categories?
Answer
Thank you for this comment. Our temptative answer is that TYG is
Did the authors include in the analysis the patients with morbid obesity?
Answer
Thank you for this question; the analysis was conducted both in patients with severe (morbid) obesity in cohort 1, and in patients with obesity in cohort 2
Did the authors identify the predictive ability of the TyG in connection with severity of obesity, a signature of concomitant CV risk factors, CVD, others?
Answer
Thank you for this question;
Other studies have shown that TyG-WHtR and TyG-WC had more robust diagnostic and / or predictive efficacy than TyG. The authors are welcome to discuss evidence in the section "Discussion".
Answer
Thank you for this question; the issue of TyG-WHtR and TyG-WC, TyG-BMI, and of TyG-BMI is discussed in the Discussion
Reviewer 2
Strengths of the Manuscript
Relevance and Novelty: The study addresses an important clinical question regarding the prognostic utility of TYG, a widely studied but inconsistently validated biomarker.
Robust Statistical Approach: The use of Cox regression analyses and Harrell’s C-index provides a solid foundation for evaluating the predictive power of TYG.
Large Sample Size: The study benefits from two well-defined cohorts, enhancing the generalizability of the findings.
Comprehensive Discussion: The authors critically compare their findings with existing literature, highlighting strengths and limitations.
Answer
Thank you for these comments
Areas for Improvement Introduction
The introduction provides a clear background on TYG and its potential role in metabolic disorders. However, a more structured explanation of the rationale behind using both an obese cohort and a general population cohort would improve readability.
Answer
Thank you for this comment. The sentences have been enriched to explain why we evaluated two cohorts.
Consider elaborating on why previous studies may have overestimated the role of TYG and how the present study aims to address these gaps.
Answer
Thank you for this comment. The current Introduction addresses the questions.
Methods
Cohort Selection: The exclusion criteria for the obese cohort (e.g., subjects undergoing bariatric surgery) are well justified. However, the justification for using NHANES data and how it complements the primary cohort should be expanded.
Answer
Thank you for this comment. The sentences have been enriched to explain why we evaluated two cohorts.
Statistical Methods: While the methods are well explained, further clarification on how missing data were handled would be beneficial.
Answer
Results
The results are well presented, with appropriate use of tables. However, there is some redundancy in the presentation of hazard ratios in the text versus the tables. Consider summarizing key findings in the text while referring to detailed tables.
Answer
Thank you for this suggestion; we have done the changes as suggested
In the obese cohort, the predictive power of TYG did not significantly improve over traditional models. This key finding should be more explicitly highlighted.
Answer
Thank you for this comment. The final sentences have been made more stronger the conclusion.
Discussion
The discussion effectively integrates findings with the literature. However, a more direct statement regarding the clinical implications of these results is needed. Should TYG continue to be used as a prognostic marker, or do the findings suggest a limited role?
Answer
Thank you for this comment. The final sentences have been made more stronger the conclusion.
The explanation for why TYG did not meaningfully enhance risk prediction is speculative. Could additional subgroup analyses (e.g., stratification by age or comorbidity burden) strengthen the argument?
Answer
Thank you for this comment. The final sentences have been made more stronger the conclusion
Conclusion
The conclusion is concise, but it should reiterate the primary clinical takeaway. If TYG is not a useful independent predictor, this should be stated unequivocally.
Answer
Thank you for this comment. The final sentences have been made more stronger conclusion
Technical and Formatting Issues
Ensure consistency in abbreviations (e.g., TYGQ vs. TYG Quartiles)
Answer
Thank you for this note. we have done the changes as suggested
Some minor typographical errors and grammatical inconsistencies should be addressed. A final proofreading is recommended.
Answer
Thank you for this note; a final proofreading was performed

Reviewer 2 Report
Comments and Suggestions for Authors
In the study, the authors found that the Triglyceride-Glucose Index did not provide additional prognostic information beyond CV risk factros and requires further validation. The findings are impressive and practically useful. The manuscript has high relevance and adequacy of the literature presented, including recognition of gaps in knowledge, interpretation of findings and significance of other recent research on the topic, and readable conclusion. However, I would like to make some comments to discuss the findings of the study.
- It remaines unclear whether the TyG are effective metrics for identifying patients with obesity at early risk of CVD to improve risk stratification or the only for other risk categories ?
- Did the authors include in the analysis the patients with morbid obesity ?
- Did the authors identify the predictive ability of the TyG in connection with severity of obesity, a signature of concomitant CV risk factors, CVD, others?
- Other studies have shown that TyG-WHtR and TyG-WC had more robust diagnostic and / or predictive efficacy than TyG. The authors are welcome to discuss evidence in the section "Discussion".
Author Response
Reviewer 1
General comments
In the study, the authors found that the Triglyceride-Glucose Index did not provide additional prognostic information beyond CV risk factors and requires further validation. The findings are impressive and practically useful. The manuscript has high relevance and adequacy of the literature presented, including recognition of gaps in knowledge, interpretation of findings and significance of other recent research on the topic, and readable conclusion. However, I would like to make some comments to discuss the findings of the study.
Answer
Thank you for these comments
Specific comments
It remains unclear whether the TyG are effective metrics for identifying patients with obesity at early risk of CVD to improve risk stratification or the only for other risk categories?
Answer
Thank you for this comment. Our temptative answer is that TYG is
Did the authors include in the analysis the patients with morbid obesity?
Answer
Thank you for this question; the analysis was conducted both in patients with severe (morbid) obesity in cohort 1, and in patients with obesity in cohort 2
Did the authors identify the predictive ability of the TyG in connection with severity of obesity, a signature of concomitant CV risk factors, CVD, others?
Answer
Thank you for this question;
Other studies have shown that TyG-WHtR and TyG-WC had more robust diagnostic and / or predictive efficacy than TyG. The authors are welcome to discuss evidence in the section "Discussion".
Answer
Thank you for this question; the issue of TyG-WHtR and TyG-WC, TyG-BMI, and of TyG-BMI is discussed in the Discussion
Reviewer 2
Strengths of the Manuscript
Relevance and Novelty: The study addresses an important clinical question regarding the prognostic utility of TYG, a widely studied but inconsistently validated biomarker.
Robust Statistical Approach: The use of Cox regression analyses and Harrell’s C-index provides a solid foundation for evaluating the predictive power of TYG.
Large Sample Size: The study benefits from two well-defined cohorts, enhancing the generalizability of the findings.
Comprehensive Discussion: The authors critically compare their findings with existing literature, highlighting strengths and limitations.
Answer
Thank you for these comments
Areas for Improvement Introduction
The introduction provides a clear background on TYG and its potential role in metabolic disorders. However, a more structured explanation of the rationale behind using both an obese cohort and a general population cohort would improve readability.
Answer
Thank you for this comment. The sentences have been enriched to explain why we evaluated two cohorts
Consider elaborating on why previous studies may have overestimated the role of TYG and how the present study aims to address these gaps.
Answer
Thank you for this comment. The current Introduction addresses the questions
Methods
Cohort Selection: The exclusion criteria for the obese cohort (e.g., subjects undergoing bariatric surgery) are well justified. However, the justification for using NHANES data and how it complements the primary cohort should be expanded.
Answer
Thank you for this comment. The sentences have been enriched to explain why we evaluated two cohorts
Statistical Methods: While the methods are well explained, further clarification on how missing data were handled would be beneficial.
Answer
Results The results are well presented, with appropriate use of tables. However, there is some redundancy in the presentation of hazard ratios in the text versus the tables. Consider summarizing key findings in the text while referring to detailed tables. Answer
Thank you for this suggestion; we have done the changes as suggested
In the obese cohort, the predictive power of TYG did not significantly improve over traditional models. This key finding should be more explicitly highlighted.
Answer
Thank you for this comment. The final sentences have been made more stronger the conclusion
Discussion
The discussion effectively integrates findings with the literature. However, a more direct statement regarding the clinical implications of these results is needed. Should TYG continue to be used as a prognostic marker, or do the findings suggest a limited role?
Answer
Thank you for this comment. The final sentences have been made more stronger the conclusion
The explanation for why TYG did not meaningfully enhance risk prediction is speculative. Could additional subgroup analyses (e.g., stratification by age or comorbidity burden) strengthen the argument?
Answer
Thank you for this comment. The final sentences have been made more stronger the conclusion
Conclusion
The conclusion is concise, but it should reiterate the primary clinical takeaway. If TYG is not a useful independent predictor, this should be stated unequivocally.
Answer
Thank you for this comment. The final sentences have been made more stronger conclusion
Technical and Formatting Issues
Ensure consistency in abbreviations (e.g., TYGQ vs. TYG Quartiles)
Answer
Thank you for this note. we have done the changes as suggested
Some minor typographical errors and grammatical inconsistencies should be addressed. A final proofreading is recommended.
Answer
Thank you for this note; a final proofreading was performed
Round 2
Reviewer 1 Report
Comments and Suggestions for Authors
The new version is ok
Author Response
Question 1
Line 53: Include the missing reference at the end of this sentence ([]).
Answer
Dear Editor, sorry for the inconvenience and thank you for the note. We have amended it, and three references have been included; the same has been done for the Discussion where we hypothesized that TYG-derived indexes behave better than TYG
Question 2
LIne 73-74: Correct the uses of () and ]: ... arterial hypertension (AH) coronary heart disease, CHD)].
Answer
Dear Editor, once again sorry for the inconvenience and thank you for the note. We have amended it.
Question 3
Line 215-216 Last sentence of the text of this manuscript needs to be revised: “The same considerations apply to < Should TYG continue to be used as a prognostic marker, or do the findings suggest a limited role?”
Answer
Thank you for your note, the final sentence now reads “Present findings suggest that both TYG and BGQ, if any, have a very limited role”
Question 4
Only need to define NHANES, BMI, once in the main text. Answer
Thank you for the Note; now the definition of NHANES appears once in the Abstract, once in the Introduction; BMI is now defined once under Material and Methods
Question 5
Need to define BGQ and TYGQ in the main text.
Answer
Thank you for the Note. BGQ and TYGQ are now defined under Material and Methods